# Data Fusion Based on an Iterative Learning Algorithm for Fault Detection in Wind Turbine Pitch Control Systems

**DOI:** 10.3390/s21248437

**Published:** 2021-12-17

**Authors:** Leonardo Acho, Gisela Pujol-Vázquez

**Affiliations:** Department of Mathematics, Universitat Politècnica de Catalunya-BarcelonaTech (ESEIAAT), 08222 Terrassa, Spain; gisela.pujol@upc.edu

**Keywords:** data fusion, iterative learning, fault detection, pitch system, wind turbines

## Abstract

In this article, we propose a recent iterative learning algorithm for sensor data fusion to detect pitch actuator failures in wind turbines. The development of this proposed approach is based on iterative learning control and Lyapunov’s theories. Numerical experiments were carried out to support our main contribution. These experiments consist of using a well-known wind turbine hydraulic pitch actuator model with some common faults, such as high oil content in the air, hydraulic leaks, and pump wear.

## 1. Introduction

Data fusion is a mathematical discipline that deals with the acquisition, processing, and combination of synergies of information gathered from sensors [1]. Data fusion can be defined as the combination of data and information from different sources, to obtain improved information [2]. This data fusion is usually done to analyze and understand a phenomenon [3,4,5], for instance a system malfunction. Data fusion techniques are present in a wide range of applications, such as smart city applications [6], allowing to manage multiple data sources; food analysis context [7]; guidance and control of autonomous vehicles [8]; medical studies [9], and so on. In addition, there are different analysis methods that combine data from different sources, where the most common options are algorithms based on optimization [10], multiblock (or multitable) methods [11], and statistical data fusion [12]. In our article, we used an original statistical parametric identification to perform data fusion, where covariance of sensory information is not required, which is generally not available.

Moreover, the data sensor has been useful to detect possible failures in the pitch actuator systems of wind turbines [13,14,15,16]. Furthermore, it turns out that a parameterized plant modeling can be a key factor for efficient fault diagnosis based on the fusion of sensor data. If in addition, only a single sensor is used for the data fusion process, an option to generate data to be merged is through an iterative process. On the other hand, it is well known that the iterative learning process can improve your performance on repetitive tasks in a finite period of time [17,18,19]. Therefore, the main objective of this article is to develop a fusion of a sensor data based on adaptive iterative learning. This process will provide data in each periodic cycle task that will be further analyzed for fault diagnosis in the wind turbine pitch actuators.

The pitch system of a wind turbine adjusts the pitch angle of the blade by turning it. In the case of a three-bladed wind turbine, there are generally three identical pitch actuators [15,20]. This part of a wind turbine is responsible for capturing the wind to convert it into mechanical energy and then into electrical one. Therefore, if the pitch actuator system has a fault, the energy efficiency conversion will be affected, among others mechanical and structural wear. Some common faults are high oil content in the air, hydraulic leaks, and pump wear. Our data fusion approach is capable of detecting these types of failures.

The iterative learning theory used for adaptive learning of a process is a key factor in many iterative learning control frameworks [17,18,19]. Therefore, an appropriate mathematical model of the pitch system will be important, and as simple as possible to perform a simple adaptive iterative learning method to our main objective. Furthermore, an iterative learning control has been used to improve control performance of proportional controllers and derivative ones [21]. Simulation results are given in [21] to support this affirmation. In [22], an iterative learning control theory is employed in a first-order hyperbolic system that helps improve controller robustness on desired time-varying trajectories. This is also supported by performing numerical examples of given hyperbolic systems. In addition, [23] shows the realization of the synchronization of non-identical neural network systems that have a variable delay in time coupled by means of an iterative learning control. According to the simulation results shown in [23], the synchronization objective is satisfied. Finally, in [24], iterative learning control is applied to a novel computational fluid dynamics model to show the performance of the controller in improving the aerodynamic load of wind turbines. In the same way, we will use numerical results to support our main contribution.

The rest of the structure of this article is as follows. Section 2 presents our data fusion approach by using a simple mathematical model of the pitch system, and the use of an adaptive iterative learning framework based on Lyapunov’s theory. Section 3 shows the results of numerical simulations and followed by Section 4, where the advantages of the proposed method are discussed. Finally, a summary is presented in Section 5.

## 2. Wind Turbine Mathematical Modeling

We use the following mathematical model of a tone actuator system [25]:(1)$$\dot{\beta }\left(t\right)=-\frac{1}{\tau }\beta \left(t\right)+\frac{1}{\tau }u_{p}\left(t\right),$$
where $u_{p}\left(t\right)$, β(t), and τ are the pitch angle command, the pitch angle, and the system time constant, respectively. This mathematical model is a simple one of the following more exact model (see [14] and references there in):(2)$$\ddot{\beta }\left(t\right)=-2\zeta \omega_{n}\dot{\beta }\left(t\right)-\omega_{n}^{2}\left(\beta \left(t\right)-u_{p}\left(t\right)\right)$$
where, once again, β(t) is the pitch angle and $u_{p}\left(t\right)$ is the pitch angle command; $\omega_{n}$ and ζ are the natural frequency and damping, respectively. Table 1 shows the healthy and faulty scenarios for a wind turbine. Therefore, our data fusion approach, for design, will use the simple model (Equation 1) and, in testing, the second model (Equation 2) under the different scenarios presented in Table 1.

## 3. Results

In this section, the statements of the iterative learning algorithm, applied to sensor data fusion, to detect pitch actuator failures, are stated.

### 3.1. Adaptive Iterative Learning Approach

The adaptive iterative learning control scheme is based on performing repetitive tasks to obtain a parameter estimation. In our case, we repeat a trajectory tracking of a wind turbine, where the unknown parameter comes from the system time constant. To do so, we rewrite system (Equation 1) as follows:(3)$$\dot{\beta }\left(t\right)=-\theta \left(\beta \left(t\right)-u_{p}\left(t\right)\right),$$
where θ=1/τ is considered an unknown parameter. Then, adaptive iterative learning deals with finding a periodic learning dynamic to observe the parameter θ that governs the pitch dynamics.

First, we have to consider the following assumptions about wind turbine pitch actuator systems:(A1)The angle pitch dynamic is limited. That is, there exists a positive constant $\beta_{M}\in R$, such that $0<\beta \left(t\right)<\beta_{M}$ for all t≥0.(A2)The systems (Equation 1) and (Equation 2) are bounded–input–bounded–output (BIBO)-stable dynamics. Hence, θ in (Equation 3) should be a positive constant parameter and assumed unknown.(A3)The pitch angle command is bounded. That is, there exists a positive constant $u_{M}\in R$, such that $|u_{p}\left(t\right)|\leq u_{M}$ for all t≥0.

We define now the next adaptive iterative learning algorithm defined over the time interval t∈[0,T]:(4)$$\begin{array}{c} \left(1-\gamma \right){\hat{\theta }}_{k}\left(t\right)=\gamma {\hat{\theta }}_{k-1}\left(t\right)+\gamma \left|\beta \left(t\right)-u_{p}\left(t\right)\right|\;,\\ {\hat{\theta }}_{k}\left(0\right)={\hat{\theta }}_{k-1}\left(T\right)\;,\\ {\hat{\theta }}_{0}\left(t\right)=\theta_{ini}\;, \end{array}$$
where *k* denotes the k-th learning cycle, or iteration number. The rest of values are: $\theta_{ini}$ is a constant parameter; 0<γ<1 is the parameter set by the user; and *T* is the time-interval of the iterative process. The above adaptive iterative dynamic is a special case to the one proposed, for instance, in [18]. Hence, this dynamic is a kind of parameter observer to θ in (Equation 3).

### 3.2. Data Fusion Design

Now, we present how to perform the data fusion of the experimental data, raw (Equation 4), obtaining for each iteration a significant information able to characterize the data. To do this, the boundedness of ${\hat{\theta }}_{k}\left(t\right)$ (Equation 4) must be established first, and then the data fusion can be accomplished.

To begin with the main result, let use define the following $L_{\infty e}$ norm [18]:(5)$${\left\|x\left(t\right)\right\|}_{\infty e}=\underset{\begin{array}{c} 0\leq t\leq T \end{array}}{\sup}\left\|x\left(t\right)\right\|,$$
where ‖·‖ denotes any vector norm. If the above norm exists, then $x\left(t\right)\in L_{\infty e}\left[0,T\right]$.

Next, we have the following result that ensures the boundedness of the iteration method (Equation 4); that is, ${\hat{\theta }}_{k}\left(t\right)$ remains in a bounded region for t∈[0,T] and for each iteration *k*.

**Theorem** **1.**
*The iterative learning process proposed in (Equation 4) linked to (Equation 3), and under the assumptions (A1)–(A3), produces $\hat{\theta }\left(t\right)\in L_{\infty e}\left[0,T\right]$ for each iteration process k=1,2,….*


The proof of Theorem 1 consists on consider an energy function related to each iteration, and show that its sequence is bounded. Then, we can ensure that the sequence of parameter ${\hat{\theta }}_{k}\left(t\right)$ is also bounded in [0,T]; that is, ${\hat{\theta }}_{k}\left(t\right)\in L_{\infty e}\left[0,T\right]$ for k=1,2,….

**Proof.** To prove the theorem, it is sufficient to see that the dynamic of (Equation 4) remains bounded. Let us define the following positive definite functional $V_{k}\left(T\right)$ as a Lyapunov-like function [18]:
(6)$$V_{k}\left(T\right)=\alpha_{1}\int_{0}^{T}{\hat{\theta }}_{k}^{2}\left(t\right)dt$$
The difference of the $V_{k}\left(t\right)$ is given by
(7)$$\begin{array}{ccc} \Delta V_{k}\left(T\right) & = & V_{k}\left(T\right)-V_{k-1}\left(T\right)\;\\ \; & = & \int_{0}^{T}\left({\hat{\theta }}_{k}^{2}\left(t\right)-{\hat{\theta }}_{k-1}^{2}\left(t\right)\right)dt \end{array}$$
Let us first simplify the integral term ${\hat{\theta }}_{k}^{2}\left(t\right)$ in (Equation 7). Hence, using (Equation 4), we get:
(8)$$\begin{array}{ccc} {\hat{\theta }}_{k}^{2}\left(t\right) & = & {\left(\frac{\gamma }{1-\gamma }\right)}^{2}{\left({\hat{\theta }}_{k-1}\left(t\right)+|\beta \left(t\right)-u_{p}\left(t\right)|\right)}^{2}\;\\ \; & = & {\left(\frac{\gamma }{1-\gamma }\right)}^{2}\left({\hat{\theta }}_{k-1}^{2}\left(t\right)+|\beta \left(t\right)-u_{p}\left(t\right){|}^{2}+2|\beta \left(t\right)-u_{p}\left(t\right)|{\hat{\theta }}_{k-1}\left(t\right)\right) \end{array}$$
Then, using (Equation 8), the difference $\Delta V_{k}$ (Equation 7) becomes:
(9)$$\begin{array}{ccc} \Delta V_{k}\left(T\right) & = & \int_{0}^{T}{\hat{\theta }}_{k}^{2}\left(t\right)-{\hat{\theta }}_{k-1}^{2}\left(t\right)\;dt\;\\ \; & = & \int_{0}^{T}\left(\frac{\gamma^{2}}{{\left(1-\gamma \right)}^{2}}-1\right){\hat{\theta }}_{k-1}^{2}\left(t\right)+\frac{\gamma^{2}}{{\left(1-\gamma \right)}^{2}}|\beta \left(t\right)-u_{p}\left(t\right){|}^{2}+\;\\ \; & \; & 2\frac{\gamma^{2}}{{\left(1-\gamma \right)}^{2}}|-\beta \left(t\right)+u_{p}\left(t\right)|{\hat{\theta }}_{k-1}\left(t\right)\;dt \end{array}$$
Now, we define γ such that $\frac{\gamma^{2}}{{\left(1-\gamma \right)}^{2}}-1\leq 0$. Then, we get:
(10)$$\begin{array}{ccc} \Delta V_{k}\left(T\right)\; & \leq & \int_{0}^{T}\frac{\gamma^{2}}{{\left(1-\gamma \right)}^{2}}|\beta \left(t\right)-u_{p}\left(t\right){|}^{2}+\;\\ \; & \; & 2\frac{\gamma^{2}}{{\left(1-\gamma \right)}^{2}}|\beta \left(t\right)-u_{p}\left(t\right)|{\hat{\theta }}_{k-1}\left(t\right)\;dt \end{array}$$
The boundedness of $V_{k}\left(T\right)$ (Equation 6) is concluded because β(t) and $u_{p}\left(t\right)$ are bounded signals. From assumptions (A1) and (A3), (Equation 10) satisfies:
(11)$$\Delta V_{k}\left(T\right)\leq \frac{\gamma^{2}}{{\left(1-\gamma \right)}^{2}}\left({\left(\beta_{M}+u_{M}\right)}^{2}T+2\left(\beta_{M}+u_{M}\right)\int_{0}^{T}{\hat{\theta }}_{k-1}\left(t\right)dt\right).$$
Finally, taking into account that ${\hat{\theta }}_{k}\left(t\right)$ is a continuous function in [0,T], for each k-th iteration, we conclude that the integral term in (Equation 11) is achievable. Hence, $\Delta V_{k}\left(t\right)<\infty$, and ${\tilde{\theta }}_{k}\left(t\right)\in L_{\infty e}$ for all *k*, and then ${\hat{\theta }}_{k}\left(t\right)\in L_{\infty e}$. So, $\hat{\theta }\left(t\right)\in C\left[0,T\right]$ for each iteration. □

Once we ensure the boundedness of the parameter estimation, we can present the data fusion scheme. The data fusion process employs experimental raw data to extract its arithmetic mean that describes it. Then, a new data raw ψ(t) is obtained with improved information. The data fusion block performs the following mathematical operation: (12)$$\psi \left(t\right)=\frac{1}{nT}\sum_{k=0}^{\left\lfloor nT\right\rceil }{\hat{\theta }}_{k}\left(t\right),t\in \left[0,T\right],$$
where ⌊nT⌉ is the nearest integer of nT, and it corresponds to the *n* iterative cycles, where each cycle is ran for t∈[0,T]. Therefore nT is the entire simulation time. Notice that all the sensor data obtained from the iterative process ${\hat{\theta }}_{k}\left(t\right)$ (Equation 4) are merged into a single data raw ψ(t). Then, this new data ψ(t) is obtained under each scenario presented in Table 1 and Table 2, and must be compared to the healthy model to establish a detection fault algorithm.

### 3.3. Fault Detection Algorithm

We now propose a diagnosis of pitch actuator failures based on the fusion data theory. Based on Theorem 1, Figure 1 shows the health monitoring system proposed for the diagnosis of failures in actuator devices in wind turbines. The data employed here are presented in Table 2 [26]. The diagnosis is based on the following steps. Under each scenario in Table 1, a data fusion raw ψ(t) (Equation 12) is obtained. Then, a decision parameter *m* is defined in each case. First, a healthy value of *m* for the nominal plant H (healthy scenario in Table 1) is derived, referred to as $m_{H}$ in Figure 1. Secondly, under each faulty case, parameter *m* is evaluated and compared to $m_{H}$ to decide if a failure occurs.

First, let us define parameter *m*: it corresponds to the regression of data fusion raw. That is, a linear relation is used to fit our data (t,ψ(t)) in (Equation 12) to a polynomial function of degree one, and by using minimum squares method. Therefore, the linear regression stage does this regression on the merged data and only *m*, the slope of the linear regression, is implemented.

Then, to detect a pitch actuator failure, the factor $r=\frac{m}{m_{H}}$ is evaluated. If r>>1, a malfunction of the system has occurred, as showed in the next section.

## 4. Numerical Simulations

Table 2 shows the stages analyzed. Therefore, the healthy model in Figure 1 refers to the *H* scenario in Table 1. The experimental parameter considered in pitch actuator exact model (Equation 2), simpler model (Equation 3) and the iterative learning algorithm (Equation 4) are defined in Table 1, Table 2 and Table 3.

For reference, the following color labeling is used: (*H*) blue, ($F_{1}$) red, ($F_{2}$) orange, and ($F_{3}$) green. By using the pitch command signal given in Figure 2, Figure 3 and Figure 4 show the results of the simulation of the proposed scheme. Then, in all simulations, additive noise is attached to the pitch command signal for the robustness analysis of the proposed method. Table 4 shows the obtained regression parameters, where in the three faulty scenarios parameter *r* is greater than 1 and the detection algorithm works. Moreover, Figure 4 pictures data fusion variable ψ(t) (Equation 4), and again the fault detection is illustrated.

Second experiment outcomes are shown in Figure 5, Figure 6 and Figure 7. Once again, Table 5 gives the reading regression parameters. From Table 1, Table 2, Table 3, Table 4 and Table 5, a threshold to the residual signal r(t) can be easily set to locate each failure. That is, despite the noise added to the data, our method is able to discern among the three different failure scenarios. As Table 5 shows, the parameter *r* for each case is located in a range of different values.

## 5. Discussion

Based on the simplest model used for the pitch actuator system (Equation 3), and because the iterative process identifies a parameter related to the system time-constant, the best option for the iterative process is to use a stepped pitch reference command, as shown in the previous simulations. However, to see the performance of our approach, we use a sine pitch command signal as shown in Figure 8. Numerical experiment results are shown in Figure 9 and Figure 10. Furthermore, Table 6 gives the related results of the iterative process results. Even in this case, the system reacts differently to different failure cases. Although the sinusoidal signal is not commonly used as a reference in estimating the time constant of a system, our approach still allows us to detect variability of this parameter. Compared to Table 5, the classification is not as robust, as expected when dealing with a sinusoidal input.Therefore, future work will be to design a residual signal, as for the sinusoidal pitch command, which will do the same job.

## 6. Conclusions

In this article, we developed an iterative learning algorithm capable of isolating pitch actuator faults based on a square pitch command signal. The option to employ the iterative learning approach is the ability to learn from the past to arrive at a present conclusion. This is an important process in system learning based on data results. Hence, our approach can be a recent contribution of this theory, to pitch actuator analyses in wind turbines.

## Figures and Tables

**Figure 1 sensors-21-08437-f001:**
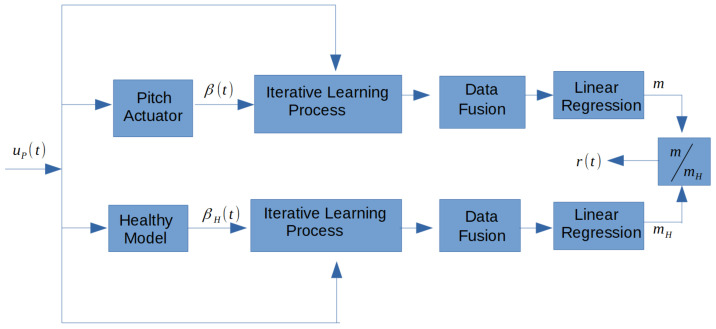
Residual signal r(t) for fault diagnosis based on an interactive learning process. The iterative learning process is as stated in Theorem 1. The linear regression block is a post-processing unit that is enabled after the end of the elapsed time for the test. Subscript *H* refers to the nominal scenario *H* in Table 1.

**Figure 2 sensors-21-08437-f002:**
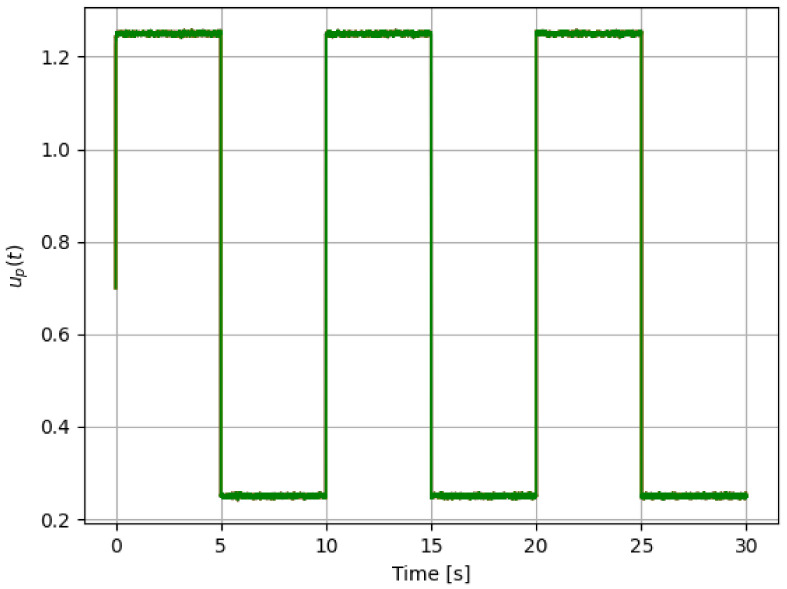
First simulation: pitch command signal for testing, where additive noise signal can be observed.

**Figure 3 sensors-21-08437-f003:**
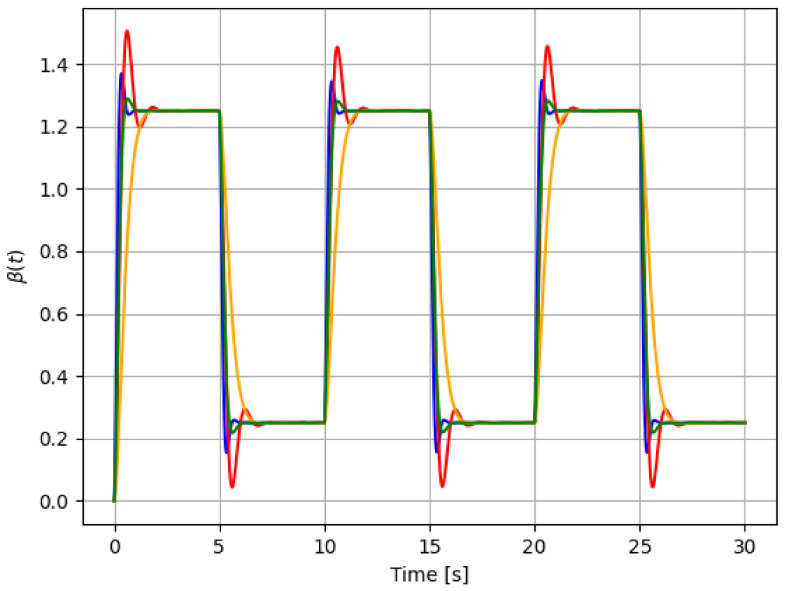
First simulation: pitch actuator responses for each case in Table 1: *H* (blue), $F_{1}$ (red), $F_{2}$ (orange), and $F_{3}$ (green). The command signal is the one in Figure 2.

**Figure 4 sensors-21-08437-f004:**
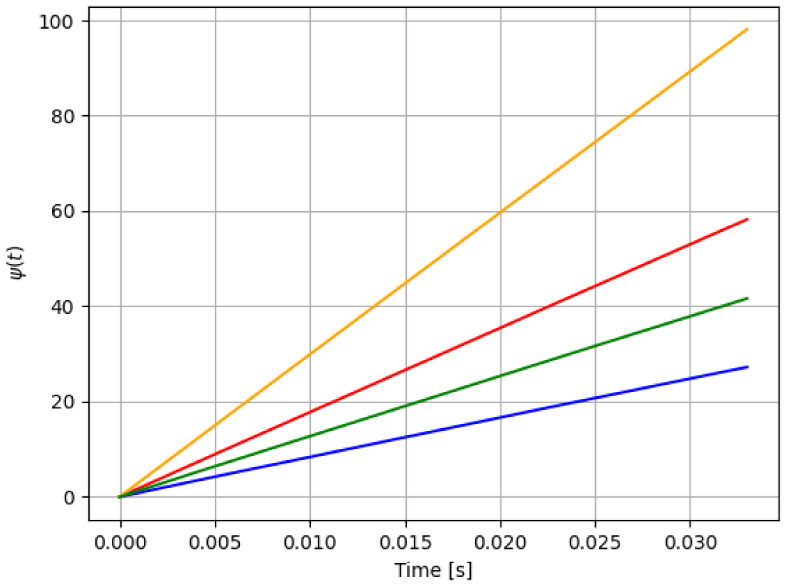
First simulation: data fusion raw ψ(t) in (Equation 4) for each case in Table 1: *H* (blue), $F_{1}$ (red), $F_{2}$ (orange), and $F_{3}$ (green). Notice that the fault detection is reached.

**Figure 5 sensors-21-08437-f005:**
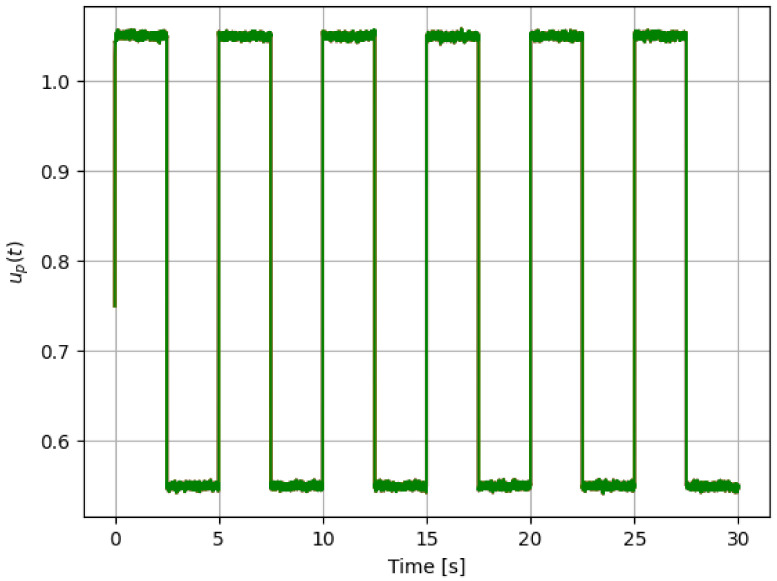
Second experiment: pitch command signal, where the additive noise is included to show the robustness of the proposed method.

**Figure 6 sensors-21-08437-f006:**
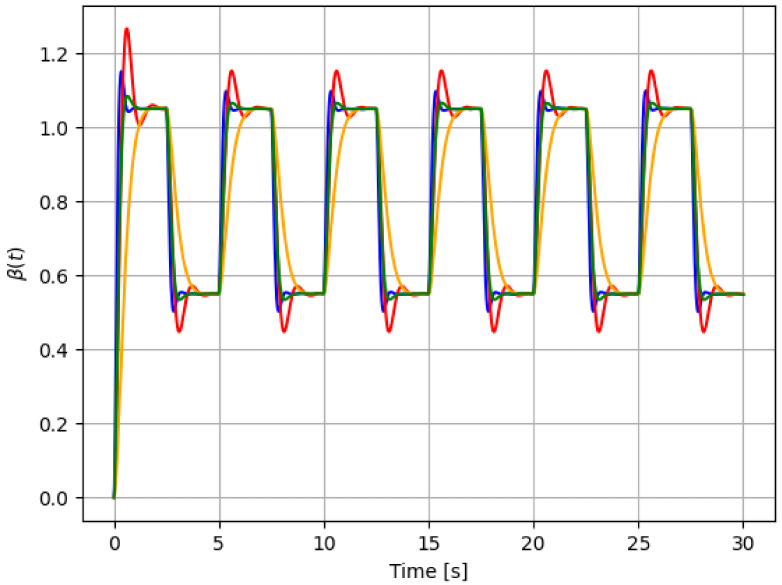
Pitch actuator responses to command in Figure 5: *H* (blue), $F_{1}$ (red), $F_{2}$ (orange), and $F_{3}$ (green).

**Figure 7 sensors-21-08437-f007:**
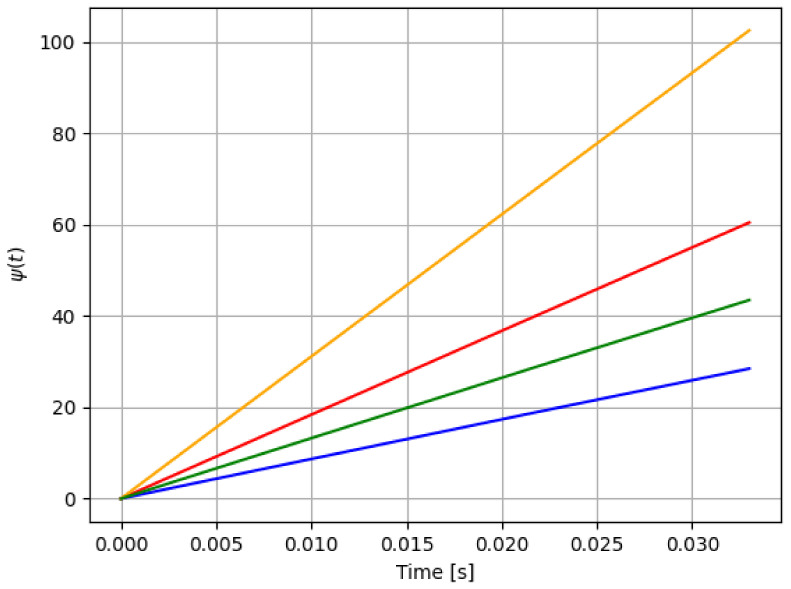
Data fusion for each case in Table 1: *H* (blue), $F_{1}$ (red), $F_{2}$ (orange), and $F_{3}$ (green), under the second experiment. This figure is related to Table 4: both indicates a pitch actuator fault detection.

**Figure 8 sensors-21-08437-f008:**
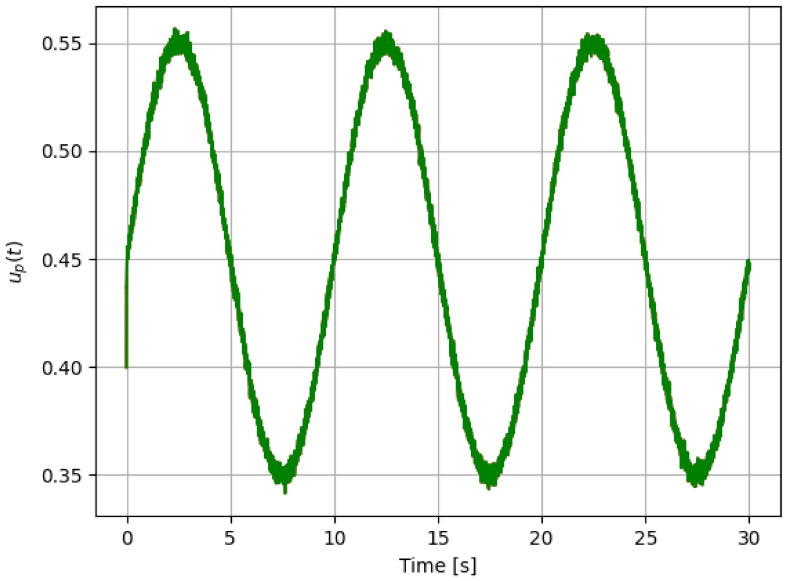
Sinusoidal pitch command signal.

**Figure 9 sensors-21-08437-f009:**
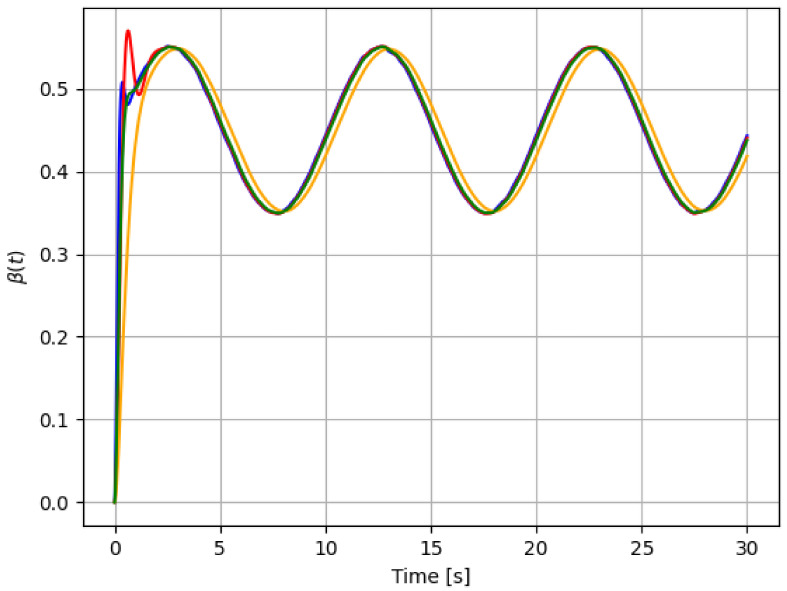
Actuator responses: *H* (blue), $F_{1}$ (red), $F_{2}$ (orange), and $F_{3}$ (green).

**Figure 10 sensors-21-08437-f010:**
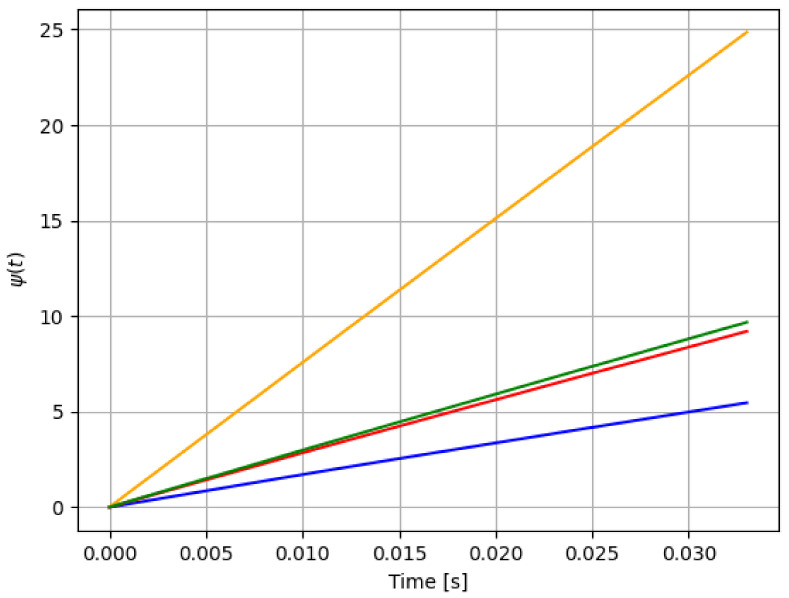
Data fusion profile: *H* (blue), $F_{1}$ (red), $F_{2}$ (orange), and $F_{3}$ (green). In this case, the fault detection is clear for $F_{2}$ faulty scenario.

**Table 1 sensors-21-08437-t001:** Common faulty scenarios [26].

Scenario	Abbreviation
No fault	*H*
High air oil content	$F_{1}$
Hydraulic leakage	$F_{2}$
Pump wear	$F_{3}$

**Table 2 sensors-21-08437-t002:** Parameters for hydraulic pitch system under common faulty scenarios [26].

Scenario	Parameter $\omega_{n}$ (rad/s)	Parameter ζ
*H*	11.11	0.6
$F_{1}$	5.73	0.45
$F_{2}$	3.42	0.9
$F_{3}$	7.27	0.74

**Table 3 sensors-21-08437-t003:** Experimental parameters.

Name	Value
γ	0.5
*T*	$\frac{1}{30}$ sec
$\theta_{ini}$	0
*n*	1000

**Table 4 sensors-21-08437-t004:** First numerical experiment results of regression parameter *m*.

Case	*m*	$r=m/m_{H}$
*H*	810.14 ($m_{H}$)	1
$F_{1}$	1732.83	2.13
$F_{2}$	2921.09	3.60
$F_{3}$	1238.93	1.52

**Table 5 sensors-21-08437-t005:** Second numerical experiment results.

Case	Regression Parameter	$r=m/m_{H}$
*H* ($m_{H}$)	848.30	1
$F_{1}$ (*m*)	1800.57	2.12
$F_{2}$ (*m*)	3053.26	3.59
$F_{3}$ (*m*)	1295.67	1.52

**Table 6 sensors-21-08437-t006:** Third numerical experiment results.

Case	Regression Parameter	$r=m/m_{H}$
*H* ($m_{H}$)	162.42	1
$F_{1}$ (*m*)	273.60	1.68
$F_{2}$ (*m*)	739.94	4.55
$F_{3}$ (*m*)	287.80	1.77

## Data Availability

Data supporting the reported results can be provided by the authors upon reasonable request.

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
