# Peer review of "Data Fusion Based on an Iterative Learning Algorithm for Fault Detection in Wind Turbine Pitch Control Systems"

_sensors, 2021, doi:10.3390/s21248437_

Round 1

Reviewer 1 Report

Overall, the manuscript presents an interesting study. Before making a final decision, the reviewer suggests the following for the authors.

  1. Paper structure should be improved, where the introduction should be separated with the mathematical model of the pitch actuator system.
  2. Table 1 should be moved to Section 3 of simulations.
  3. Please explain more about the linear regression of data fusion at the end of page 4, lines 120-122.

Author Response

Thank you very much for your inputs. Please see the PDF attached file.

Reviewer 2 Report

 In order to improve the quality of this paper, the following comments are suggested to authors:

Major Comments:

  1. The reported works discuss on use of data fusion but efficiency of individual data is not discussed. If we really want to know the efficiency of fusion data, then it should be compared with the data analysis from individual sources. A comparative study to show the impact of data fusion should be included.
  2. Many new advanced techniques developed and reported on fault detection of wind turbine pitch control system based on multiclass optimal margin distribution machine or, based on large margin distribution machine optimized by the state transition algorithm or, based on improved deep belief network with SCADA data etc.

To prove the superiority of suggested approach a comparative remark or related studies should be included.

  1. Require more extensive literature survey.
  2. Numerical simulations are not sufficient to establish a new algorithm; some experimentation should be included. Some graphs are included as experimentation but I didn’t find any experimentation details in this article. In my view numerical simulation should not be tagged as experimentation.

Minor Comments:

  1. It is observed that uniformity is missing to represent figures. Somewhere it is written “Figure.”, somewhere it is figure. In my view it should not be. It will reduce the standards of the publications.
  2. Similar type of plots should be in the same scale otherwise proper changes should not be observed. For example, Figures 4 & 7 plotted in y axis from 0 to 100 whereas similar type Figure 10 plotted in y axis from 0 to 25. If any special reason for this can you explain.

Author Response

(The authors gave the same response as above.)

Round 2

Reviewer 2 Report

may be accepted in the current form